# Quality of the Wind Wave Forecast in the Black Sea including Storm Wave Analysis

**Stanislav Myslenkov [1,2,3,*], Alexander Zelenko [3], Yuriy Resnyanskii [3], Victor Arkhipkin [1] and Ksenia Silvestrova [2]**

[1] Department of Oceanology, Lomonosov Moscow State University, 119991 Moscow, Russia; victor.arkhipkin@gmail.com
[2] Shirshov Institute of Oceanology RAS, 117997 Moscow, Russia; ksberry@mail.ru
[3] Hydrometeorological Research Centre of the Russian Federation, 123242 Moscow, Russia; zelenko@mecom.ru (A.Z.); resn@mecom.ru (Y.R.)
**\*** Correspondence: stasocean@gmail.com

**Abstract:** This paper presents the results of wind wave forecasts for the Black Sea. Three different versions utilized were utilized: the WAVEWATCH III model with GFS 0.25 forcing on a regular grid, the WAVEWATCH III model with COSMO-RU07 forcing on a regular grid, and the SWAN model with COSMO-RU07 forcing on an unstructured grid. AltiKa satellite altimeter data were used to assess the quality of wind and wave forecasts for the period from 1 April to 31 December 2017. Wave height and wind speed forecast data were obtained with a lead time of up to 72 h. The presented models provide an adequate forecast in terms of modern wave modeling (a correlation coefficient of 0.8–0.9 and an RMSE of 0.25–0.3 m) when all statistics were analyzed. A clear improvement in the wave forecast quality with the high-resolution wind forecast COSMO-RU07 was not registered. The bias error did not exceed 0.5 m in an SWH range from 0 to 3 m. However, the bias sharply increased to −2 or −3 m for an SWH range of 3–4 m. Wave forecast quality assessments were conducted for several storm cases.

**Keywords:** Black Sea; wave modeling; wind wave forecast; significant wave height; WAVEWATCH III; SWAN; GFS; COSMO-RU07

## 1. Introduction

Information regarding wind waves in the sea is the most requested parameter for many industries. Active oil and mineral field exploration and the development of transport and tourism occur in this region, according to reports of the International Centre for Black Sea Studies [1]. The Black Sea is an area of intensive navigation. It has many uses, including tourist and cultural activities, marine fishing, cargo transport, and rescue operations, to name a few. The study of the hydrometeorological conditions of the Black Sea is relevant due to the active economic development of this region. Much research in recent years has been focused on the Black Sea wave climate [2–6]. The central problem for wave modelling is the forecast. Wind wave conditions are limiting factors for economic activity and development of infrastructure in the coastal zone. Storm waves can destroy infrastructure in coastal zones as well as offshore, cause economic damage, and can threaten human life. Most of the accidents that caused oil spills occurred in stormy weather. A higher quality of wave forecasts will certainly contribute to the sustainable development of the region. For example, the oil spill in the Kerch Strait in 2007 was partially caused by sea river vessels that were caught in a storm in the Black Sea [7]. For such ships, wave heights of more than 4–5 m are already dangerous.

There has been rapid development in the theory and practice of using wave energy converters [8]. A wave forecast often includes wave energy parameters. Thus, forecasts are useful for planning the location of wave energy converters and power generators.

Furthermore, wave forecasts could help in protecting the converters from waves that are too big. Research related to the use of wave energy undoubtedly contributes to the sustainable development of water resources of the world's oceans. Therefore, qualitative wave forecasts provide possible benefits for industries in the Black Sea region.

The Hydrometeorological Center of Russia has developed and launched the operation of a wind wave forecast system for the world's oceans and Russian seas [9]. Wave forecasts based on a spectral wave model are monitored by WAVEWATCH III (WW3) [10]. WAVEWATCH III was developed by the National Center for Environmental Prediction (NCEP) based on WAVEWATCH I and WAVEWATCH II (the Delft University of Technology and NASA Goddard Space Flight Center, respectively). The model employed meteorological data from two sources: (1) The semi-Lagrangian atmospheric SL-AV model operating online in the Hydrometeorological Center of Russia [11]; and (2) the output products of the Global Forecasting System (GFS) of NCEP [12,13]. Estimates of the quality of wind wave forecasts made by this system are given in [14] for the world's oceans; in [15] for the Azov, Black, and Caspian Seas; and in [16] for the Baltic Sea. This forecast system was recommended by the Roshydromet Central Methodological Committee for Hydrometeorological and Heliogeophysical Forecasts to be the main system for monitoring the world's oceans as well as individual sea basins.

The new system of wind wave forecasts in the Black Sea, utilizing a high spatial resolution in nearshore zones, was launched in 2016 [17]. This system is based on the Simulating Waves Nearshore (SWAN) spectral model. SWAN is a third-generation wave model, developed at the Delft University of Technology [18]. In contrast to WAVEWATCH III, this implementation works on an unstructured grid. Such a grid allows one to obtain wave parameters with high spatial resolution in certain areas.

Unfortunately, most articles concerning the description and quality of wave forecasting systems in Russia have been published in highly specialized journals in the Russian language and access is therefore limited to Russian speakers. The purpose of this study is to describe three operative forecast systems, in English, and draw certain comparisons between them.

The quality of wind wave forecasts largely depends on the quality of the input meteorological information, primarily wind speed data. As the forecast time interval increases, there is an increasing dependency between the errors in wave height forecasts and the errors in wind speed forecasts [15]. Improving the quality of meteorological forecasts can be achieved by increasing the spatial resolution, and in particular, using products of mesoscale atmospheric models, such as the COSMO-RU07 model [19]. This non-hydrostatic, limited-area, atmospheric model was developed and supported by the Consortium for small-scale modeling (COSMO). Consequently, RU07 represents its correspondence to the Russian domain and the 7 km spatial resolution. In each case, however, it is necessary to evaluate the quality of the wave forecast. A higher resolution of wind fields does not always lead to an improvement in the wave modeling results [20].

There are several works devoted to the hindcast and forecast of wind waves in the Black Sea. In previous work [21], a wave hindcast system for the Black Sea based on Weather Research and Forecasting (WRF) and SWAN models is presented. The wave-height quality estimates (root mean square error (RMSE) = 0.27–0.31 m and correlation coefficient (R) = 0.88–0.91) were obtained in comparison with the observational data on the Gloria platform.

The most complete error analysis of retrospective wave modeling with various reanalysis in the Black Sea is given in [22]. In this work, wind fields from Climate Forecast System Reanalysis (NCEP/CFSR), ERA Interim, JRA, Merra, ERA40, and European Center for Medium-Range Weather Forecasts (ECMWF) reanalysis were used for wave hindcast. By comparing the results of wave modeling with the observational data, it was found that the highest quality was achieved using the NCEP/CFSR reanalysis (RMSE = 0.32 m, R = 0.88). However, the wind speed from NCEP/CFSR reanalysis data was overestimated according to satellite observation data with an average systematic error of 1 m/s. The

authors decided to correct the reanalysis with the spatial wind correction following the satellite data. However, the quality of wave simulation slightly decreased. Thus, it is an indicator of the variability in time and space of the Bias error of the wind speed or the presence of such error in the satellite data.

In the work [23], the wave forecast system for the Black Sea is based on the SKIRON forecasts and SWAN model. The wave heights quality estimates in comparison with the satellite data provide RMSE = 0.39 m (0–1 days forecast) and 0.49 (2–3 days forecast).

There are several more wind wave forecast systems for the Black Sea, but there are no estimates of their quality in open sources [24–27].

This article presents a comparison of the wave heights forecast for different lead times with satellite observation data. The authors attempt to determine the quality of forecasts obtained with three forecast systems: WAVEWATCH III with GFS 0.25 on a regular grid, the WAVEWATCH III with COSMO-RU07 on a regular grid, and the SWAN with COSMO-RU07 on an unstructured grid.

## 2. Materials and Methods

### 2.1. Wave Forecast Systems

The main wind wave forecast system for the Black and Azov Seas, which has been in development at the Hydrometeorological Center of Russia since 2010, is based on the WAVEWATCH III v 3.14 spectral model with WAM4 wind parameterization and dissipation with a set of BAJ parameters [10]. For the Black and Azov Seas, forecasts are calculated as part of a single constructed multi-grid with a spatial grid spacing of 6.0′ × 6.0′ (about ~4 × 7 km) in the Black Sea and 1.2′ × 1.2′ (about 0.7 × 1.5 km) in the Azov Sea (Figure 1). The spectral resolution of the model is 24 directions (Dq = 15°), and a frequency range includes 25 intervals (from 0.042 to 0.41 Hz).

Wind speed forecast data are used from the GFS system with a spatial resolution of ~0.1° and time step 1 h. The time step of wave model integration was 270 s for the Black Sea and 90 s for the Sea of Azov.

This system provides wave forecast every day for up to 5 days with time step 3 h. The following characteristics of the wind waves are calculated: significant wave height (SWH), propagation direction, average wave length, and average wave period. The initial conditions for each forecast were set according to the wave spectra data from the previous day's forecast. It is also possible to spin up the model from calm conditions according to the analysis of preceding wind forecast fields.

The second additional version of the wave forecast system is based on the COSMO-RU07 wind speed forecast. COSMO-RU07 is a mesoscale atmospheric model with a high spatial resolution of ~7 km and time step 1 h [19]. This atmospheric model is based on a system of thermohydrodynamic equations describing a compressible air flow in a humid atmosphere and is based on a system of equations describing the basic laws of continuum mechanics: the law of conservation of mass, the law of conservation of momentum (momentum), the law of conservation of energy, and the equation of state (Clapeyron's equation). The results of numerical weather forecast using the COSMO-Ru system are received daily 4 times a day according to the initial data at 0, 6, 12, and 18 h UTC, and they are prepared and sent to users on servers, for example, in the form of GRIB files. The grid not only includes the European part of Russia but also the Urals, part of Western Siberia, and almost all of Europe region. A more detailed description of COSMO-RU07 is given in [19,28]. All other WAVEWATCH III wave model settings were the same.

A more detailed description of the WW3 model, computational grid, and quality assessment are given in [15,16].

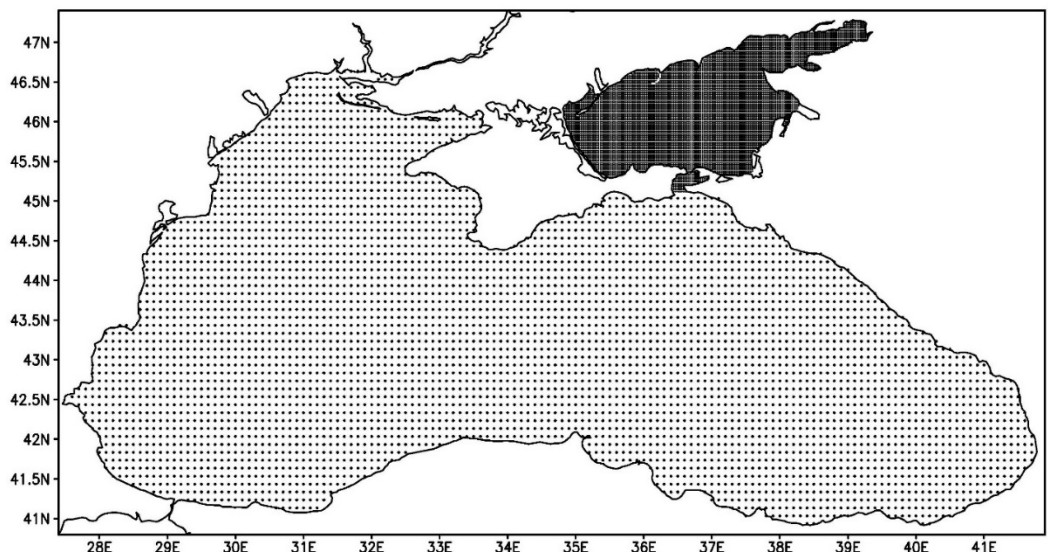

**Figure 1.** Regular computational grid of WAVEWATCH III model for the Black Sea.

The third version of the wind wave forecast system with high spatial resolution in nearshore zones is based on the SWAN 41.01 spectral wave model [18,29]. Calculations are carried out on an unstructured computational grid, which includes the Black and Azov Seas with a spatial resolution of 6–12 km in the open sea. Several areas: Kerch Strait, Tsemes Bay, and the Sochi region are calculated with higher resolution up to 200 m. The total number of grid nodes is 12,131 (Figure 2). Due to the special unstructured grid, this system is more suitable for predicting waves in coastal and shallow areas.

The SWAN model start in the "GEN3" mode with listed parameters: KOMEN growth scheme (cds2 = $2.36 \times 10^{-5}$, stpm = $3.02 \times 10^{-3}$), three- and four-wave interaction (Quadruple, Triad), wave breaking (Breaking constant, alfa = 1.0, gamma = 0.73), and bottom friction (Friction Jonswap Constant cf = 0.067). The spectral resolution of the model is 72 directions (Dq = 5°), and a the frequency range includes 37 intervals (from 0.03 to 1.0 Hz). The time step of the wave model integration is 900 s.

This version of the wave forecast system is based on the COSMO-RU07 wind speed forecast with a spatial resolution of ~7 km and a time step of 1 h [19].

This system provides wave forecast every day for up to 3 days with a time step every 3 h. The following characteristics are calculated: SWH, propagation direction, average wave length, and average wave period.

The wave model starts 48 h before the first forecast from calm conditions, i.e., sufficient initial wave conditions are provided.

A more detailed description of the model, computational grid, and quality assessment are given in [20,30,31].

Thus, the results of the wind wave forecast based on 3 different versions of wave models are presented: the WAVEWATCH III model with GFS forcing on a regular grid (WW3-GFS), the WAVEWATCH III model with COSMO forcing on a regular grid (WW3-Cosmo), and the SWAN model with COSMO forcing on an unstructured grid (SWAN-Cosmo).

*2.2. Satellite Data*

Satellite data from the AltiKa altimeter aboard the SARAL satellite were used for the quality assessment of wind and wave forecasts on deep water. The SWH and wind speed data have a spatial resolution of about 7 km along the track and are available on the RADS database [32]. The root mean square error (RMSE) of SWH from different altimeters in comparison to observational data is usually estimated as 0.3 m [33]. The quality of AltiKa altimeter data is estimated by experts higher than the quality of Envisat, Jason 1, 2 data.

The quality assessment results of AltiKa according to the wave buoys data show that RMSE is 0.21 m and systematic error (Bias) is 0.04 m for the SWH. For the wind speed, RMSE and Bias are 1.75 m/s and 0.25 m/s, respectively [34].

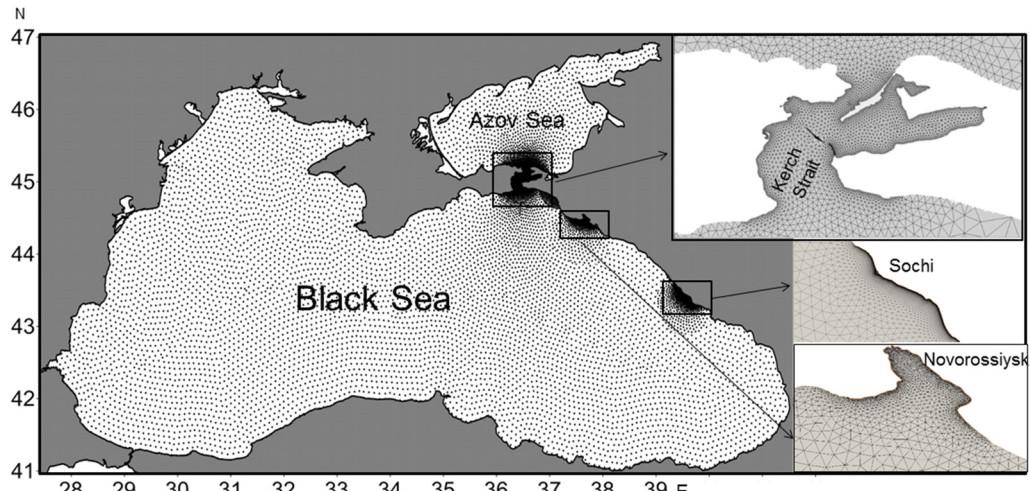

**Figure 2.** Unstructured computational grid for the SWAN wave model system with high spatial resolution in nearshore zones of the Black Sea.

In this paper, AltiKa satellite data are compared with modeled data for the period from 1 April to 31 December 2017. In total, about 14,000 values of SWH and wind speed were obtained from satellite measurements. The altimeter track map and point density are shown in Figure 3.

Data with bad quality flags or closer than 10 km to the shore were filtered (deleted) from the analysis.

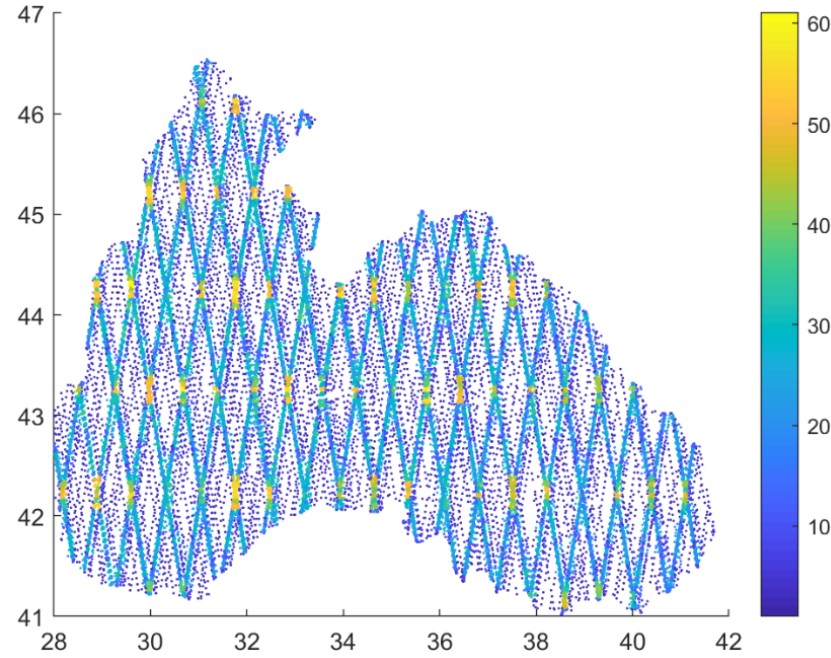

**Figure 3.** AltiKa altimeter tracks and point density for the period from 1 April to 31 December 2017.

The distance between the compared points was less than 8 km. Since the satellite moves over the Black Sea mainly at ~3:00 and ~17:00 UTC, the comparison was made for the forecast lead time at 3 h, 15 h (for 1-day forecast), 27 h, 39 h (for 2-day forecast), 51 h,

and 63 h (for 3-day forecast). There were about 7000 values of satellite data for each time after "closest to forecast" point searching.

The configurations of wave forecast systems have the output time step of 3 h. Therefore, the time difference between compared model and satellite data was ±1.5 h.

The model quality assessments is based on the standard statistical parameters:

$$\text{Bias} = \sum_{i=1}^{N} \frac{1}{N}(P_i - O_i) \tag{1}$$

$$\text{RMSE} = \sqrt{\frac{1}{N-1}\sum_{i=1}^{N}(P_i - O_i)^2} \tag{2}$$

$$\text{SI} = \frac{\text{RMSE}}{\frac{1}{N}\sum_{i=1}^{N} O_i} \tag{3}$$

$$R = \frac{\sum_{i=1}^{N}((P_i - \overline{P})(O_i - \overline{O}))}{\sqrt{\left(\sum_{i=1}^{N}(P_i - \overline{P})^2\right)\left(\sum_{i=1}^{N}(O_i - \overline{O})^2\right)}} \tag{4}$$

where N is the total number of data, $P_i$ is the forecast value, $O_i$ is the observed value, $\overline{P}$ is the mean forecast value, and $\overline{O}$ is the mean observed value.

## 3. Results

There were three versions of wave height and wind speed forecasts: WW3-GFS, WW3-COSMO, and SWAN-COSMO. The forecasts were obtained for each day with a lead time of up to 72 h for the period from 1 April to 31 December 2017. The data array selection is about 700 points per month for each forecast or about 7000 points of one forecast time interval for the entire test period.

### 3.1. Wind Quality Assessment

The quality assessment of wind forecast fields was carried out. The wind speed according to GFS and COSMO-RU07 data was compared with satellite data.

Estimates of the quality of wind forecasts are obtained for different lead times. The comparison was made with wind speed data interpolated on the wave models' grid. In a perfect case, the results of WW3-COSMO and SWAN-COSMO should match, because the original wind data were the same. The observed differences could be caused by interpolation procedures or by different output time steps.

Figure 4 is an example of comparing the wind speed from COSMO-RU07 with a lead time of 15 h and satellite data. This scatter plot shows good agreement between the wind forecast and satellite data. The correlation coefficient is 0.82, and the RMSE is 1.8 m/s. The highest density of points is concentrated in the range of 3–4 m/s. There is a large spread in the high speed range (more than 10 m/s) and the forecast values slightly overestimate the satellite data.

The scatter cloud becomes larger for a lead time of 39 h, the correlation coefficient decreases to 0.79, and RMSE increases to 2 m/s (Figure 5).

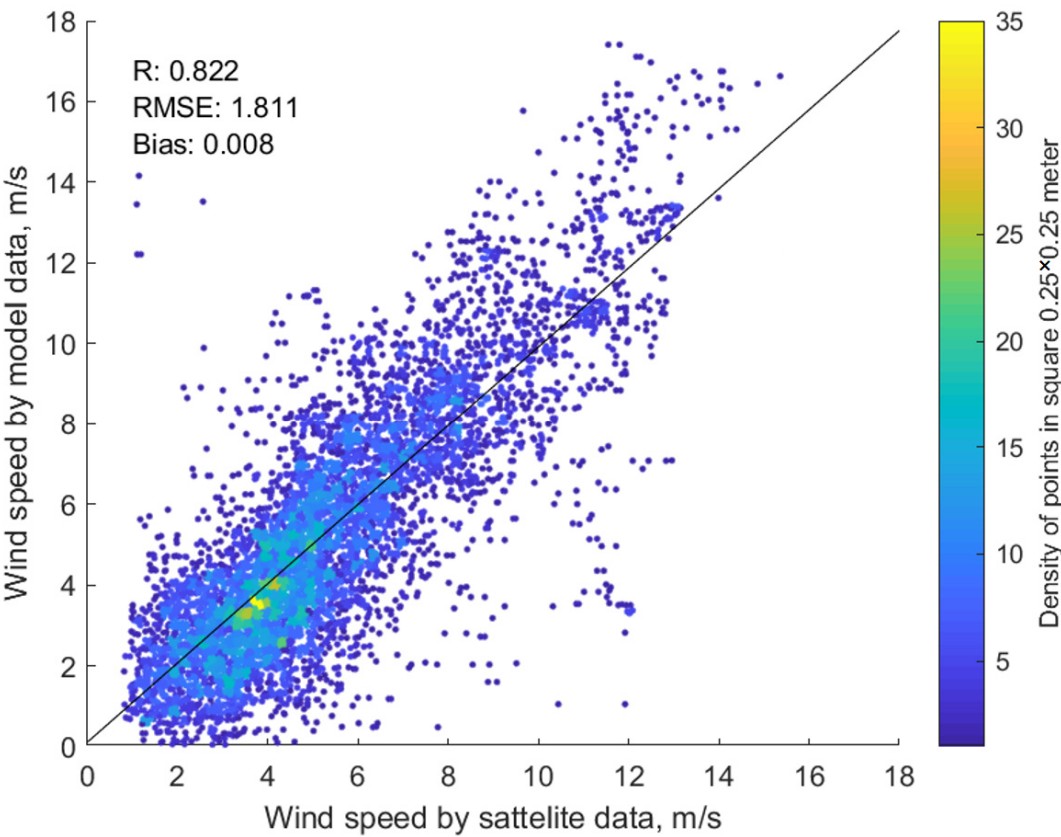

**Figure 4.** Scatter plot for wind speed from COSMO-RU07 with a lead time of 15 h and satellite data.

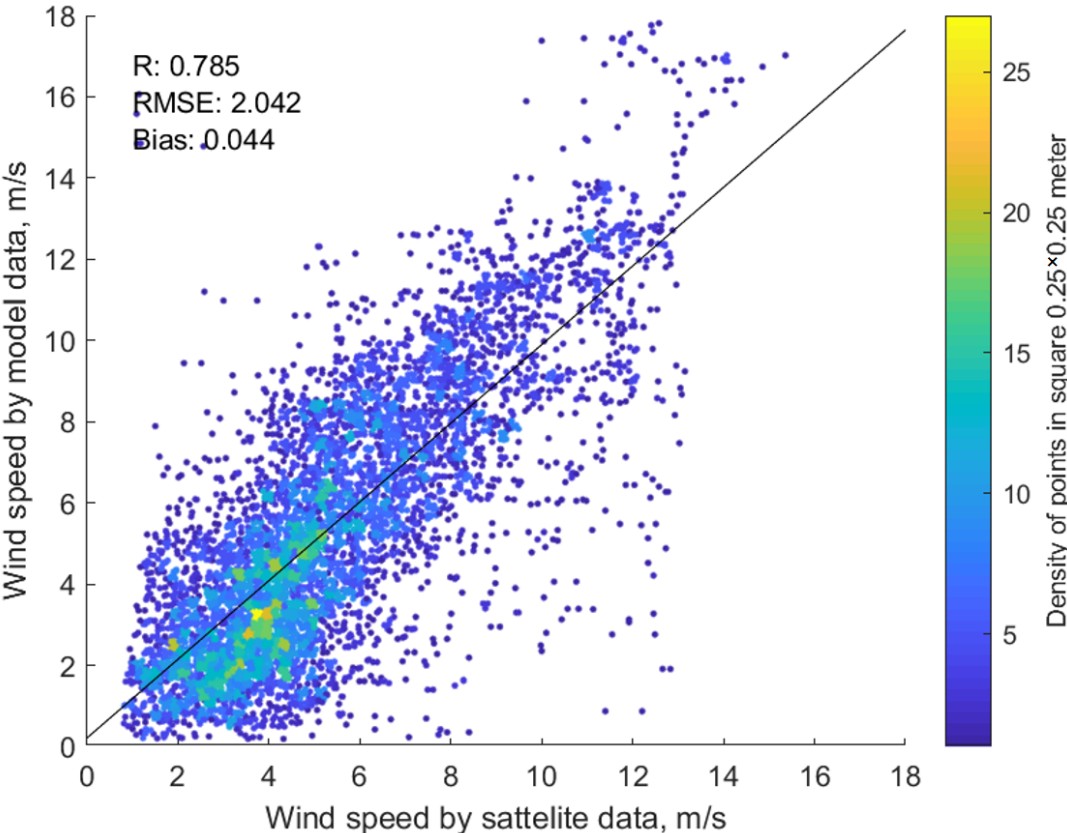

**Figure 5.** Scatter plot for wind speed from COSMO-RU07 with a lead time of 39 h and satellite data.

Detailed statistical information on the quality of wind forecasts is given in Table 1. These estimates are obtained separately for each forecast lead time for the entire test period (from April to December 2017). The correlation coefficient decreases when the forecast goes from the first day up to the third. The highest correlation coefficient for lead time 3–39 h is for the WW3-COSMO model. However, for the WW3-GFS version, the forecast is more stable for a lead time of 51–63 h. The SWAN-COSMO version, which should completely coincide with WW3-COSMO, shows a slightly worse result, which is related, we believe, to a lower spatial resolution of this version of the wave model. According to the RMSE, the best result is shown by the WW3-GFS version for almost all periods of lead time. According to a Bias error, the WW3-GFS version stably overestimates wind speed by 0.3–0.7 m/s, and the WW3-COSMO and SWAN-COSMO versions show multidirectional errors of about +0.14/−0.25 m/s. Furthermore, Table 1 contains information about the SI, which shows the ratio of the RMSE and average wind speed. SI changed from 0.32–0.33 to 0.44–0.45 for different lead times. It is quite difficult to draw conclusions about better forecasts according to the results shown in Table 1; all models give very similar results.

**Table 1.** Statistical estimates of the wind speed forecast quality for the entire test period for different forecast lead times.

| Forecast Lead Time | R (Correlation Coef) | | | RMSE | | | Bias | | | SI | | |
|---|---|---|---|---|---|---|---|---|---|---|---|---|
| | WW3-GFS | WW3-COSMO | SWAN-COSMO | WW3-GFS | WW3-COSMO | SWAN-COSMO | WW3-GFS | WW3-COSMO | SWAN-COSMO | WW3-GFS | WW3-COSMO | SWAN-COSMO |
| 3 | **0.83** | **0.83** | 0.82 | **1.63** | 1.65 | 1.68 | 0.33 | 0.14 | 0.08 | 0.32 | 0.33 | 0.33 |
| 15 | 0.80 | 0.82 | 0.80 | 1.73 | 1.79 | 1.82 | 0.36 | **0.00** | −0.07 | 0.33 | 0.34 | 0.34 |
| 27 | 0.76 | 0.80 | 0.78 | 1.90 | 1.80 | 1.83 | 0.44 | 0.12 | 0.10 | 0.38 | 0.36 | 0.35 |
| 39 | 0.73 | 0.78 | 0.74 | 1.98 | 2.03 | 2.17 | 0.30 | 0.05 | −0.14 | 0.38 | 0.38 | 0.40 |
| 51 | 0.71 | 0.70 | 0.69 | 2.21 | 2.21 | 2.21 | **0.73** | 0.12 | −0.01 | 0.44 | 0.43 | 0.42 |
| 63 | 0.73 | 0.70 | **0.68** | 2.02 | 2.37 | **2.40** | 0.33 | -0.17 | −0.25 | 0.38 | 0.44 | 0.45 |

Next, consider the results of assessing the quality of wind speed forecasts in different seasons of the year. Table 2 shows the correlation coefficients for each month for different lead times. The correlation coefficient on the first day for all versions of forecasts is high (0.8–0.9) for April, August, September, October, and December. In June, July, and November, this correlation is lower and equal to 0.5–0.7 (Figure 6). The data array for each month is not enough to provide statistically significant results, but the main features in seasonal change were noted. The quality of the wind speed forecast certainly depends on the synoptic situation; with weak and unstable winds and with a high influence of local mesoscale processes, the quality of the forecasts should decrease. We cannot say that the COSMO-RU07 mesoscale model provides a better quality in the summer months. However, we can see clearly that the R is more than 0.8 when the wind speed is higher than 5.5–6 m/s (Figure 6).

**Table 2.** R (correlation coefficients) when comparing wind speed forecasts and satellite data separately for each month for different forecast lead times.

| | Forecast Lead Time, h | April | May | June | July | August | September | October | November | December |
|---|---|---|---|---|---|---|---|---|---|---|
| WW3-GFS | 3 | 0.89 | 0.77 | 0.70 | 0.79 | 0.86 | 0.86 | 0.86 | 0.69 | 0.80 |
| | 15 | 0.79 | 0.65 | 0.73 | 0.88 | 0.84 | 0.86 | 0.78 | 0.74 | 0.83 |
| | 27 | 0.83 | 0.72 | 0.66 | 0.70 | 0.73 | 0.82 | 0.79 | 0.45 | 0.75 |
| | 39 | 0.55 | 0.48 | 0.73 | 0.78 | 0.68 | 0.85 | 0.77 | 0.73 | 0.84 |
| | 51 | 0.83 | 0.69 | 0.69 | 0.58 | 0.66 | 0.75 | 0.69 | 0.37 | 0.73 |
| | 63 | 0.69 | 0.53 | 0.48 | 0.78 | 0.65 | 0.81 | 0.79 | 0.65 | 0.80 |
| SWAN-COSMO | 3 | 0.92 | 0.88 | 0.68 | 0.54 | 0.84 | 0.85 | 0.89 | 0.68 | 0.78 |
| | 15 | 0.87 | 0.75 | 0.72 | 0.89 | 0.88 | 0.81 | 0.64 | 0.74 | 0.81 |
| | 27 | 0.80 | 0.84 | 0.64 | 0.73 | 0.75 | 0.83 | 0.79 | 0.56 | 0.79 |
| | 39 | 0.65 | 0.68 | 0.64 | 0.72 | 0.74 | 0.80 | 0.75 | 0.72 | 0.82 |
| | 51 | 0.64 | 0.62 | 0.69 | 0.82 | 0.67 | 0.81 | 0.65 | 0.44 | 0.67 |
| | 63 | 0.74 | 0.45 | 0.46 | 0.69 | 0.51 | 0.74 | 0.60 | 0.52 | 0.73 |
| WW3-COSMO | 3 | 0.91 | 0.78 | 0.69 | 0.75 | 0.85 | 0.85 | 0.89 | 0.66 | 0.78 |
| | 15 | 0.89 | 0.66 | 0.73 | 0.87 | 0.84 | 0.81 | 0.77 | 0.73 | 0.82 |
| | 27 | 0.79 | 0.83 | 0.64 | 0.64 | 0.78 | 0.84 | 0.84 | 0.62 | 0.80 |
| | 39 | 0.73 | 0.60 | 0.61 | 0.85 | 0.72 | 0.79 | 0.84 | 0.72 | 0.85 |
| | 51 | 0.63 | 0.65 | 0.67 | 0.79 | 0.68 | 0.81 | 0.68 | 0.42 | 0.67 |
| | 63 | 0.74 | 0.52 | **0.43** | 0.72 | 0.50 | 0.73 | 0.69 | 0.58 | 0.73 |

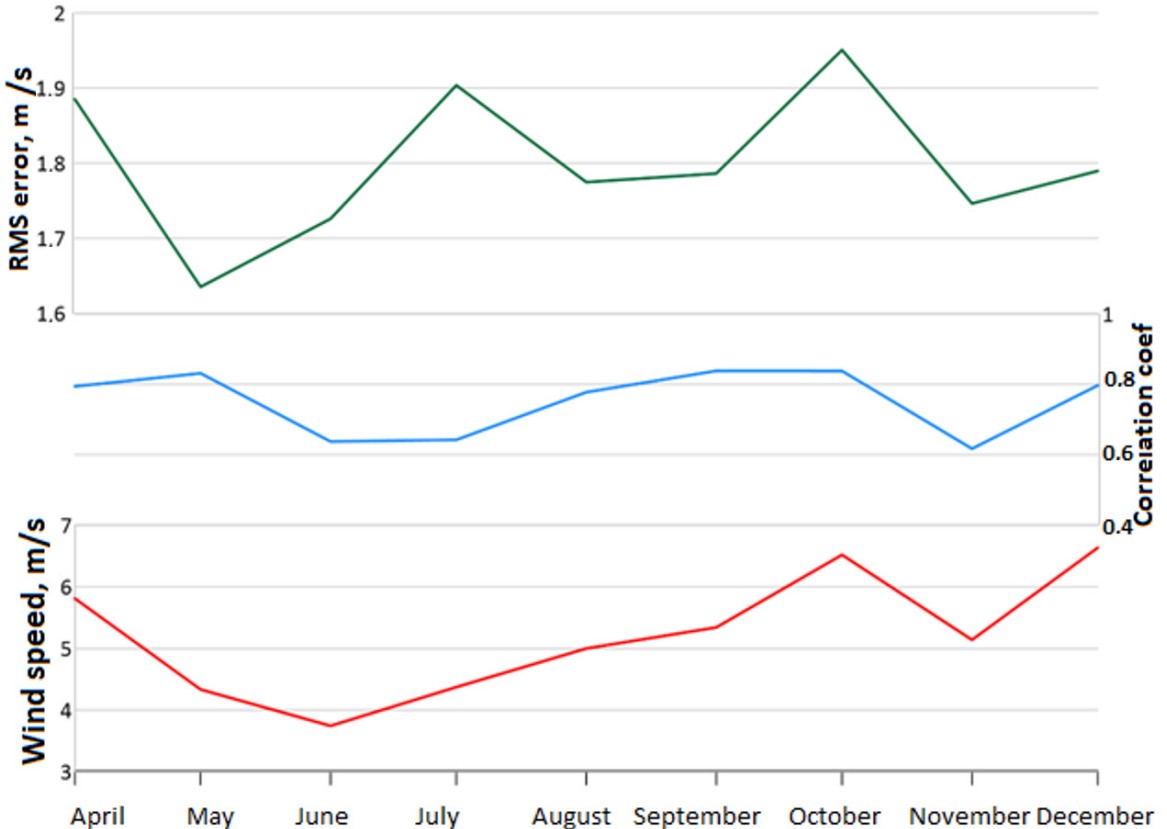

**Figure 6.** Seasonal variability of wind speed, correlation coefficient, and RMSE for the forecast interval time of 27 h according to the version of WW3-COSMO.

### 3.2. *Wave Quality Assessment*

At the second stage of the study, the quality of SWH forecasts for different lead times was evaluated. Comparison of the calculated SWH with satellite data was carried out for the assessments of wind speed forecasts.

An example of the comparison between SWH from WW3-COSMO and satellite data with a lead time of 15 h is shown in Figure 7. The scatterplot presents the good agreement of forecasts and satellite data. The correlation coefficient is 0.92, and the RMSE is 0.28 m. The areas in the scatter plot with the highest point density are concentrated near the bisector. In the upper range of wave heights (more than 4 m), the forecast slightly overestimates the satellite data. However, the number of such cases is small. Statistically, Bias is still negative (–0.12 m); therefore, the model underestimates the SWH.

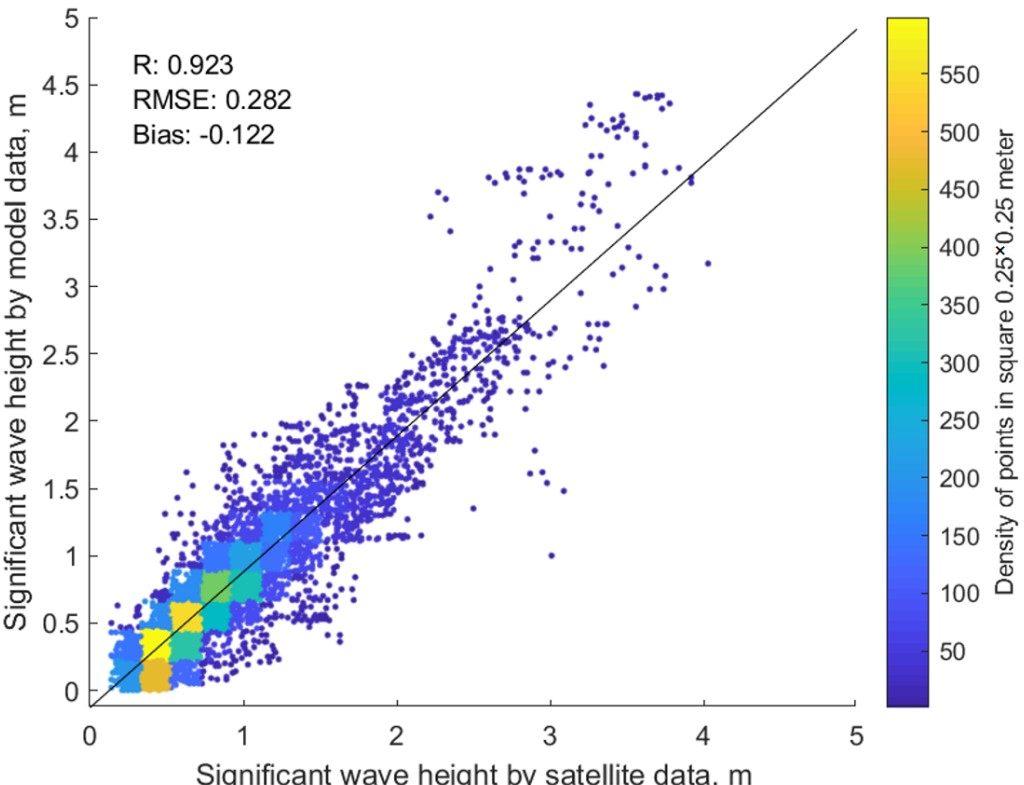

**Figure 7.** Scatter plot for significant wave heights for a forecast time interval of 15 h according to WW3-COSMO.

For the forecast lead time of 39 h, the scatter plot is similar; however, the scatter cloud increases and the correlation coefficient falls to 0.85 (Figure 8.). We can see two areas outside the main cloud of points. The first area is where the SHW forecast is ~5 m and SWH satellite is ~3.5–4 m, and a second area is where the SHW forecast is ~1.5 m and SWH satellite is ~3.5–4 m. Most likely these areas depict separate storm events where the model overestimates or underestimates the SWH.

For other versions of forecast system, the picture is approximately the same as for WW3-COSMO.

Statistical estimates of the quality of the wave height forecast for each lead time for the entire test period (from April to December 2017) are given in Table 3. The correlation coefficient is ~0.9 for all versions of the forecast before lead time touches 15 h and then decreases. The highest correlation coefficient was for the WW3-COSMO model. Since the SWAN-COSMO version showed a result which is less accurate than other versions in terms of wind speed, wave height results also turned out the worst. For RMSE and Bias errors, the best result is shown by the WW3-GFS version for almost all lead times. For all

versions of the forecast, the Bias error is small but negative. Therefore, the models slightly underestimate the wave heights. It should be noted that the WW3-GFS has a lower correlation coefficient than WW3-COSMO, but it gives better results in RMSE and Bias errors. SI changed from 0.29–0.31 to 0.43–0.47 for different lead times. WW3-GFS has a lower SI for all lead times.

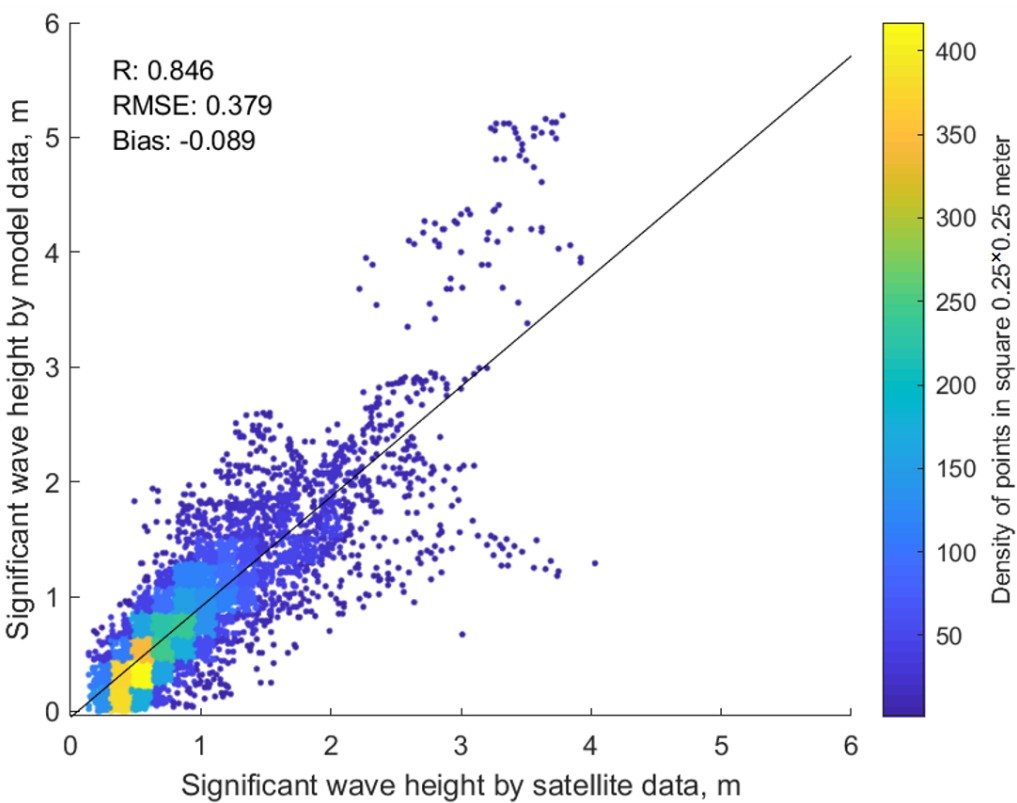

**Figure 8.** Scatter plot for significant wave heights for a forecast time interval of 39 h according to WW3-COSMO.

**Table 3.** Statistical estimates of the SWH forecast quality for the entire test period for different forecast lead times.

| Forecast Lead Time | R (Correlation Coef) | | | RMSE | | | Bias | | | SI | | |
|---|---|---|---|---|---|---|---|---|---|---|---|---|
| | WW3-GFS | WW3-COSMO | SWAN-COSMO | WW3-GFS | WW3-COSMO | SWAN-COSMO | WW3-GFS | WW3-COSMO | SWAN-COSMO | WW3-GFS | WW3-COSMO | SWAN-COSMO |
| 3 | 0.88 | 0.88 | 0.84 | **0.27** | 0.30 | 0.35 | −0.09 | −0.12 | **−0.18** | 0.31 | 0.34 | 0.39 |
| 15 | 0.91 | **0.92** | 0.89 | **0.27** | 0.28 | 0.32 | −0.09 | −0.13 | −0.17 | 0.29 | 0.31 | 0.35 |
| 27 | 0.85 | 0.85 | 0.80 | 0.30 | 0.32 | 0.35 | −0.07 | −0.12 | −0.14 | 0.34 | 0.37 | 0.39 |
| 39 | 0.84 | 0.85 | 0.79 | 0.33 | 0.37 | 0.41 | −0.10 | −0.09 | −0.16 | 0.35 | 0.39 | 0.43 |
| 51 | 0.81 | 0.76 | **0.73** | 0.35 | 0.41 | 0.42 | **−0.04** | −0.11 | −0.15 | 0.40 | 0.46 | 0.46 |
| 63 | 0.77 | 0.78 | 0.76 | 0.41 | 0.44 | 0.44 | −0.10 | −0.11 | −0.17 | 0.43 | 0.47 | 0.46 |

The correlation coefficient in assessing the quality of the SWH is higher than in assessing the wind speed. This happens because wind waves have less spatial variability than the wind field and waves accumulate wind energy from a larger area. Therefore, the

total amount of wind energy in meteorological forecasts is close to reality. Consequently, the wave modeling tends to be successful.

The results of assessing the quality of wave height forecasts for each month are shown in Table 4. A low correlation coefficient is observed only in June and August when there were low wind speeds and weak wind waves. The highest correlation coefficient for all versions of forecasts is observed in September. For example, for the WW3-GFS, the correlation coefficient varies from 0.95 to 0.88 for all lead times (Figure 9). As was noted above, the correlation coefficients for SWH were higher than for wind speed. This is also true for the results for each month, except for June, when the correlation coefficient for the wind speed was 0.7 according to the WW3-GFS is and was 0.64 according to the SWH. The same results were obtained for other versions of forecasts. The data array for each month is not large enough to provide statistically significant results.

In general, wind speed and wave height forecasts were considered successful, because the correlation coefficients are high, and the RMSE is about 0.2–0.3 m. In the summer months, the models give higher errors due to the weak winds but the SWH is small.

**Table 4.** R (correlation coefficients) when comparing SWH forecasts and satellite data separately for each month for different forecast lead times**.**

| | Forecast Lead Time/Month | April | May | June | July | August | September | October | November | December |
|---|---|---|---|---|---|---|---|---|---|---|
| **WW3-GFS** | 3 | **0.93** | 0.82 | 0.64 | 0.88 | 0.77 | **0.95** | 0.87 | 0.84 | 0.84 |
| | 15 | 0.91 | 0.79 | 0.82 | 0.94 | 0.85 | **0.95** | 0.92 | 0.85 | 0.92 |
| | 27 | 0.91 | 0.78 | 0.63 | 0.85 | 0.68 | 0.93 | 0.87 | 0.64 | 0.86 |
| | 39 | 0.75 | 0.73 | 0.80 | 0.86 | 0.71 | 0.93 | 0.91 | 0.84 | 0.90 |
| | 51 | 0.87 | 0.75 | 0.56 | 0.80 | 0.62 | 0.90 | 0.85 | 0.48 | 0.78 |
| | 63 | 0.56 | 0.62 | 0.54 | 0.85 | 0.69 | 0.88 | 0.83 | 0.71 | 0.76 |
| **SWAN-COSMO** | 3 | 0.90 | 0.85 | 0.60 | 0.84 | 0.74 | 0.85 | 0.86 | 0.83 | 0.79 |
| | 15 | 0.90 | 0.89 | 0.74 | 0.87 | 0.85 | 0.91 | 0.92 | 0.77 | 0.90 |
| | 27 | 0.84 | 0.84 | 0.47 | 0.80 | 0.61 | 0.81 | 0.85 | 0.64 | 0.84 |
| | 39 | 0.72 | 0.86 | 0.60 | 0.69 | 0.70 | 0.86 | 0.89 | 0.67 | 0.84 |
| | 51 | 0.67 | 0.74 | **0.41** | 0.84 | 0.50 | 0.81 | 0.75 | 0.47 | 0.81 |
| | 63 | 0.71 | 0.75 | 0.58 | 0.66 | 0.63 | 0.80 | 0.81 | 0.58 | 0.86 |
| **WW3-COSMO** | 3 | 0.91 | 0.83 | 0.45 | 0.84 | 0.63 | 0.94 | 0.93 | 0.89 | 0.88 |
| | 15 | **0.95** | 0.75 | 0.81 | 0.94 | 0.89 | **0.97** | 0.92 | 0.90 | 0.92 |
| | 27 | 0.84 | 0.87 | 0.42 | 0.81 | 0.61 | 0.93 | 0.87 | 0.79 | 0.90 |
| | 39 | 0.75 | 0.80 | 0.70 | 0.87 | 0.80 | 0.90 | 0.92 | 0.81 | 0.87 |
| | 51 | 0.65 | 0.79 | 0.36 | 0.83 | 0.46 | 0.91 | 0.77 | 0.57 | 0.84 |
| | 63 | 0.68 | 0.71 | 0.66 | 0.76 | 0.75 | 0.80 | 0.79 | 0.67 | 0.81 |

Fisher′s transformation algorithm was used to assess the significance of the correlation coefficients. The results of estimates for the WW3-GFS version are given in Table 5. Confidence limits of 95% were very close to the original R (<0.01) due to the large length of the series and the relatively low deviation. Correlation analysis of SWH forecasts for each month with the length of the series around ~600 points reveals that 95% confidence limits were ±0.2–0.3 to the original R.

According to correlation analysis, results for the entire test period were better than for the separate months. Such a difference was predictable in line with the data series length. Nevertheless, all correlation coefficients are significant.

**Table 5.** Fisher's transformation of the SWH correlation coefficient for the entire test period for the WW3-GFS version.

| Forecast Lead Time | R (Sample Correlation Coef) | N | Fisher's z | Standard Deviation $S_r$ | 95% Confidence Limits | |
|---|---|---|---|---|---|---|
| | | | | | Left | Right |
| 3 | 0.876 | 7145 | 1.359 | 0.012 | 0.871 | 0.881 |
| 15 | 0.905 | 6427 | 1.499 | 0.012 | 0.901 | 0.909 |
| 27 | 0.853 | 6505 | 1.267 | 0.012 | 0.846 | 0.859 |
| 39 | 0.842 | 6357 | 1.227 | 0.013 | 0.834 | 0.849 |
| 51 | 0.812 | 6506 | 1.132 | 0.012 | 0.803 | 0.820 |
| 63 | 0.765 | 6410 | 1.008 | 0.012 | 0.754 | 0.775 |

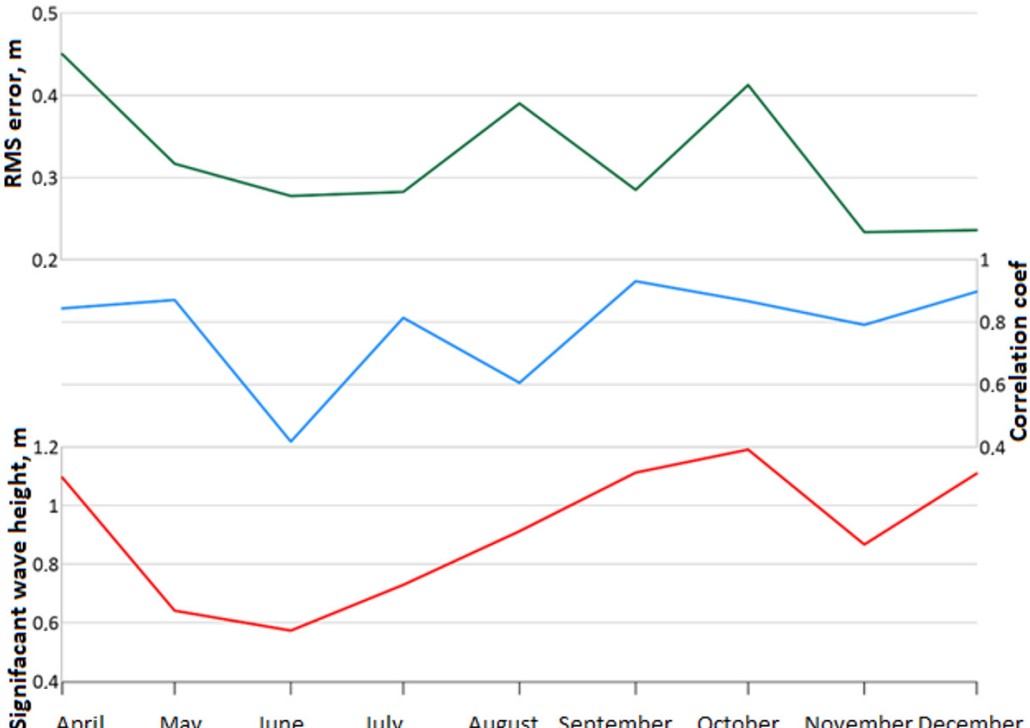

**Figure 9.** Seasonal variability of average SWH, R correlation coefficient, and RMS error for the forecast lead time of 27 h according to the version of WW3-COSMO.

Earlier estimates for the WW3-GFS version showed the following results: a correlation coefficient of 0.8–0.87 and RMSE of 0.36–0.44 m [15]. Our estimates are slightly better because only AltiKa satellite data were used. AltiKa has better quality than Envisat or ERS2.

If we compare the obtained results with the assessments of the quality of other forecast systems for the Black Sea [21,23], the presented versions have the same quality.

It should be noted that the distribution of the Bias error in space is not uniform. Figure 10 shows Bias error distribution over the Black Sea according to the WW3-COSMO and WW3-GFS versions. We can see that Bias error was around ±0.1 m for forecast lead times of 15 h over the Black Sea, except the several coastal regions, where local winds probably have an influence on the wind waves. The greatest Bias error are observed in the region of Novorossiysk, where Novorossiysk bora periodically acts. The Bias error increases to −0.2 m for forecast lead times of 39 h in the several parts of the open sea.

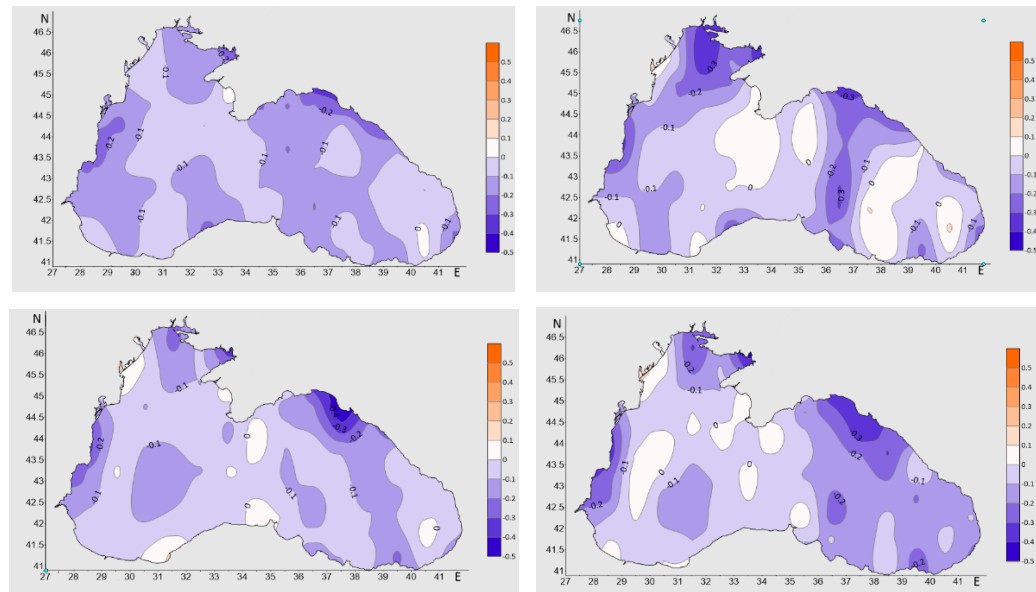

**Figure 10.** Bias error distribution (m) of SWH in the Black Sea for the wave forecast according to the WW3-COSMO for forecast lead times of 15 h and 39 h (**upper panel**); according to the WW3-GFS for lead times of 15 h and 39 h (**bottom panel**).

*3.3. Wave Forecasts Quality Assessments for a Different Wave Height*

General quality assessments of SWH forecasts do not provide correct information about the possible errors for high waves in stormy conditions. The wave height is usually very small in the Black Sea. SWH greater than 3 m was obtained in 1–2% of total cases [3]. Situations when the SWH is more than 2 m can be considered as a storm. Figure 11 shows the distribution function for SWH from observations for the entire 2017 year (total sample length ~7100). The SWH varies mostly from 0 to 2 m, while there are only a few cases when the SWH is more than 3.5 m. Consequently, in general statistics, estimates for wave heights of 0–2 m are prevailing, while we are more interested in storm waves forecast.

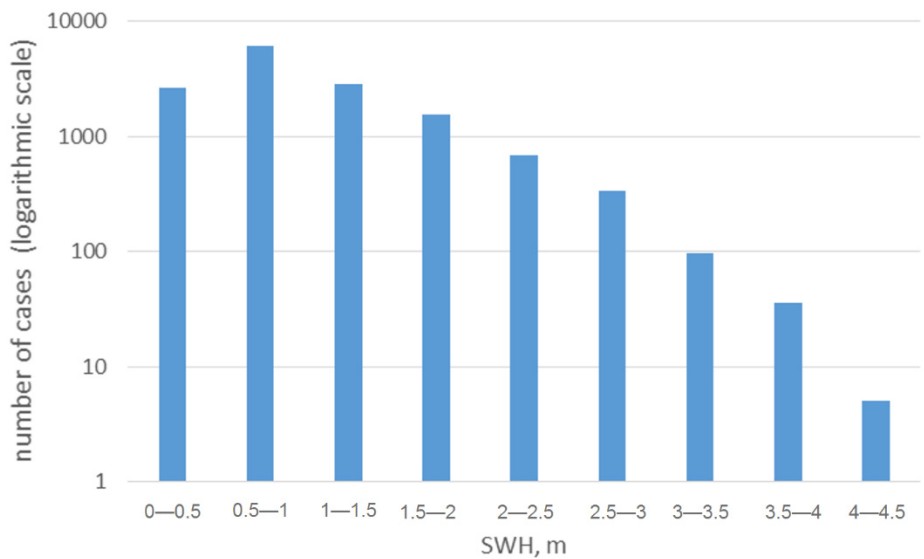

**Figure 11.** Distribution function—number of cases for a different SWH range by satellite data in 2017.

To solve this problem, we have considered the distribution of the Bias for the different SWH range. Figure 12 shows the distribution of the Bias for different lead times

for different versions of the forecast system in 2017. The BIAS does not exceed 0.5 m in the SWH range from 0 to 3 m. However, the Bias sharply increases to −2 or −3 m for the SWH range, and to 3–4 m for some forecast lead times. The negative Bias indicates an underestimation of forecasts. However, statistical estimates for SWH more than 3.5 m have only a ~5–10 cases (Figure 12). Therefore, only 1–2 storm events are predicted either well or badly, so the result is appropriate. For the GFS-WW3, the errors for wave heights above 3 m do not exceed 2.5 m (the smallest value for three versions). The GFS-WW3 also has low RMSE values for all lead times. This is also confirmed for stormy conditions and we can conclude that the GFS-WW3 version is better than the other. However, only the statistical results for SWH less 3 m can be considered correct and have enough sample.

### 3.4. Wave Forecasts Quality Assessments in the Storm Cases

There were two particularly severe storms according to satellite data in 2017. The first storm occurred on 19 April 2017. The WW3-GFS, with a lead time of 27 h from 18 April, calculated that the SWH was 4.5 m in the central part of the sea (Figure 13). The satellite altimeter tracked in the western part of the sea. The results of comparison between the along-track data and WW3-GFS different lead times data are shown in Figure 14. All forecast versions underestimate the wave height along the altimeter track significantly. The maximum wave height was about 4.1 m in altimetry data and was 2.6 m according to the forecast. The results of the forecasts for different lead times are not very close to the measurements; however, a tendency for an error decrease with a decrease in lead time is observed.

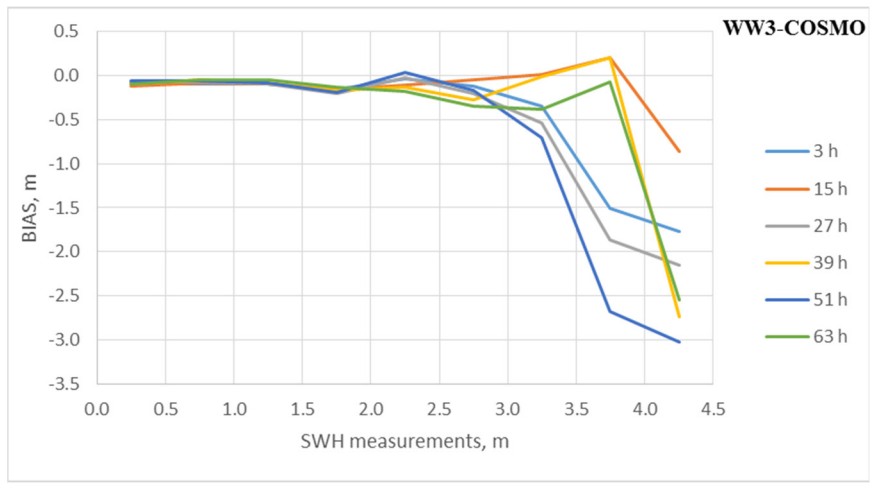

(**a**)

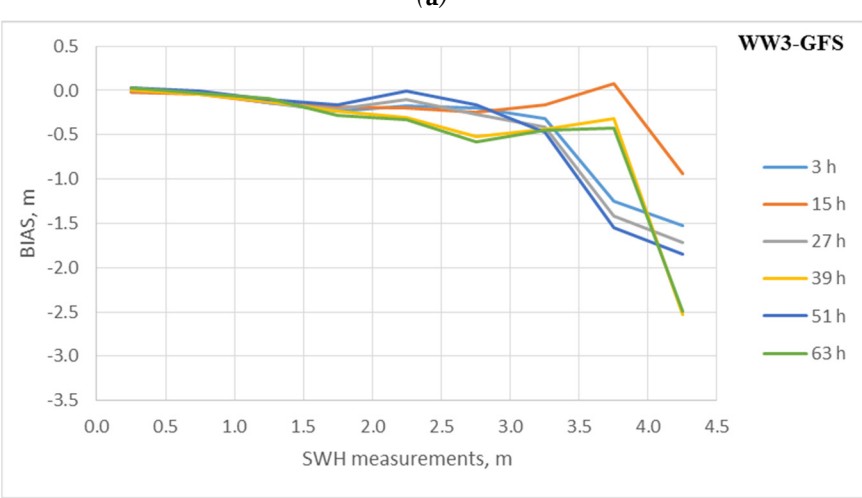

(**b**)

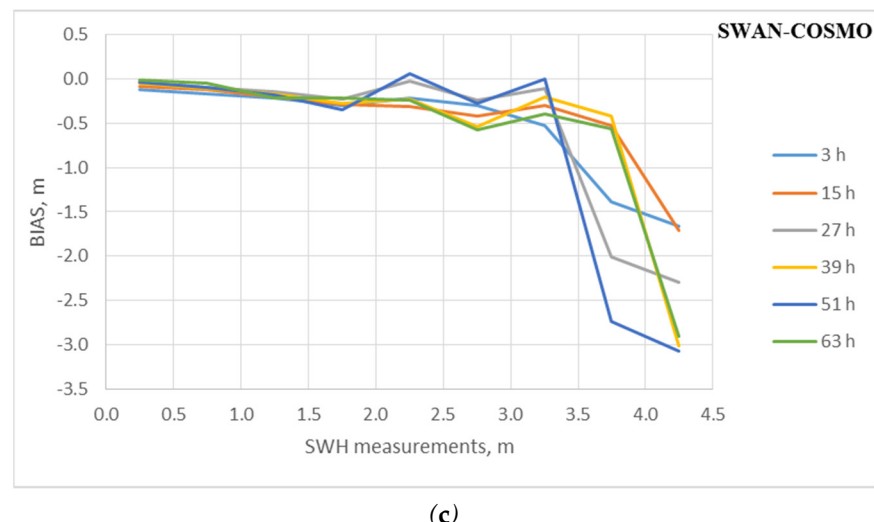

(**c**)

**Figure 12.** Distribution of Bias for a different SWH range based on WW3-COSMO (**a**), WW3-GFS (**b**), and SWAN-COSMO (**c**).

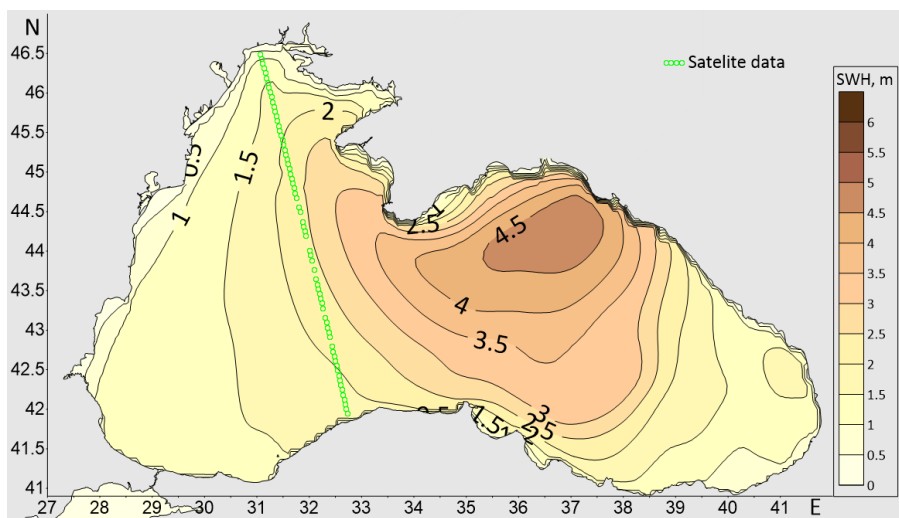

**Figure 13.** Map of SWH based on WW3-GFS and forecast time of 27 h from 18 April 2017. Green dotes marked the satellite altimetry track.

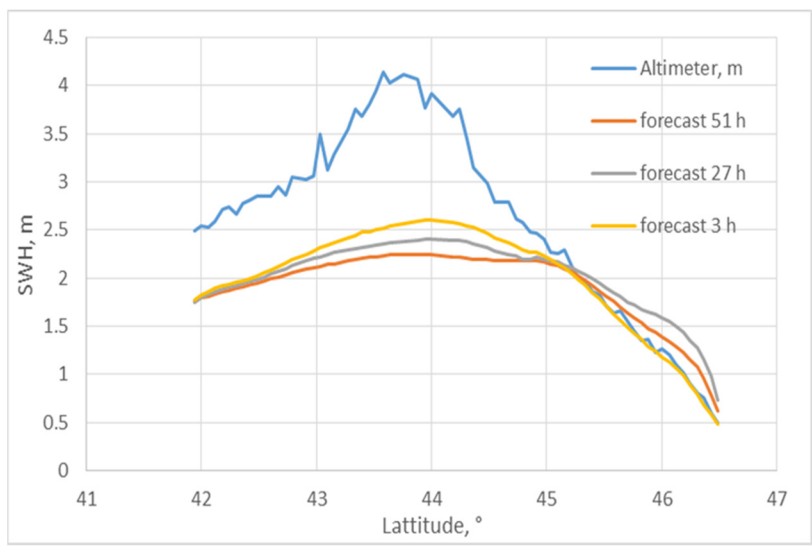

(**a**)

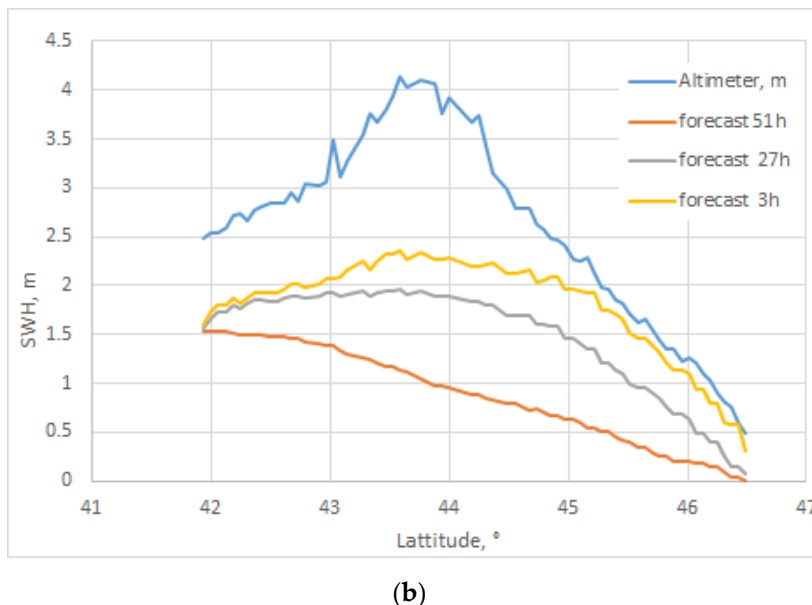

(**b**)

**Figure 14.** SWH along altimetry track 19 April 2017 and SWH based on WW3-GFS (**a**) and WW3-COSMO (**b**) for different forecast lead times.

The second storm occurred on 30 October 2017. The WW3-GFS, for the lead time of 63 h, shows that SWH reached 5.5 m in the eastern central part of the sea (Figure 15). The altimeter tracked in the middle part of the sea and almost crossed the zone of maximum waves. The results of comparison between satellite data and WW3-GFS data for different lead times are shown in Figure 16. All forecast options significantly overestimate the SWH. According to the altimeter data, the maximum wave height was about 3.7 m and, according to the forecast data, the SWH maximum reached 4.5 m. The modeled results do not match the measurements closely for different lead times.

In total, there are 5–7 cases of such storms in the Black Sea per year; however, altimeter tracks do not always intercept zones of strong wave development. Thus, the statistical series for high waves contains very few values and it make a great problem for quality estimates.

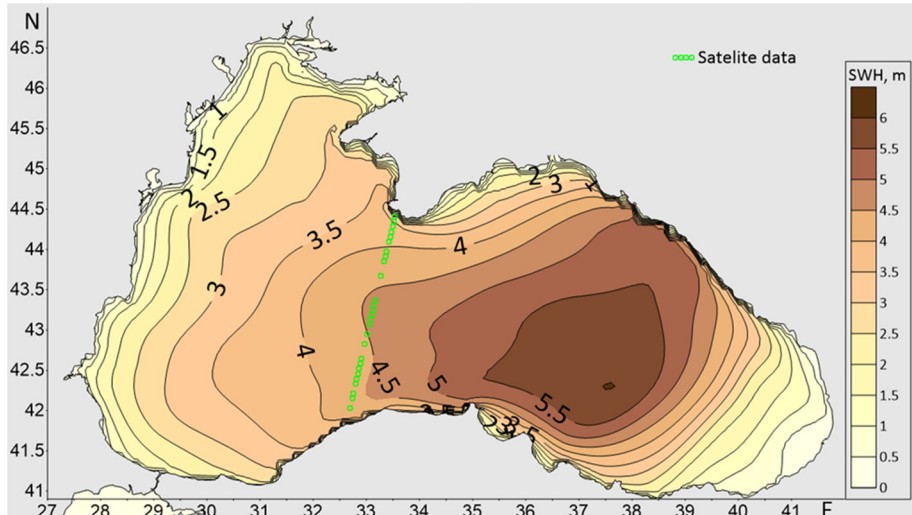

**Figure 15.** Map of SWH based on WW3-GFS for lead time 63 h from 28 October 2017. Green dotes marked the satellite altimetry track.

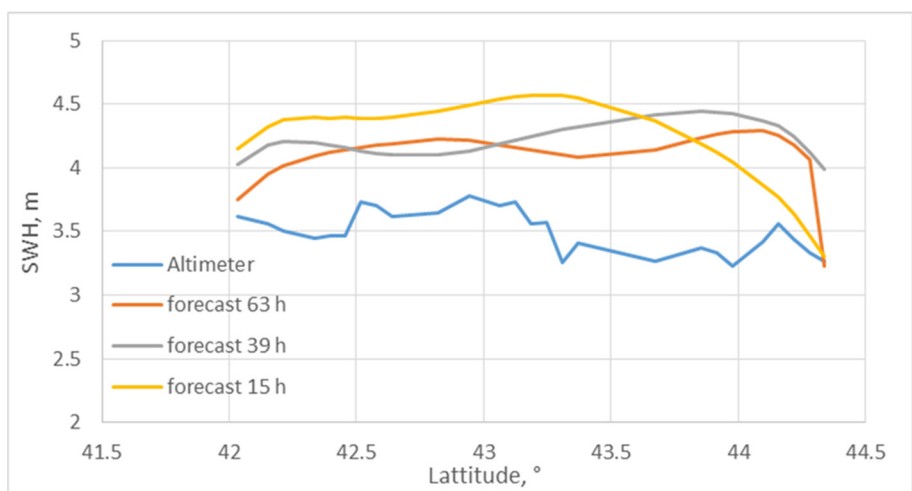

**Figure 16.** SWH along altimetry track 30 October and SWH based on WW3-GFS for different forecast lead times.

## 4. Discussion

Statistical analysis showed the following results for the WW3-GFS version: R = 0.91–0.76 (from 1 to 3 day of forecast), RMSE = 0.27–0.41 m, Bias = ~−0.1 m, and SI = 0.29–0.43. In general, our results are comparable to modern investigations in wave forecast quality assessments [35,36]. WW3-GFS version has a slightly lower R and larger SI, caused by a small average of SWH. In the open ocean with an average SWH of 3–4 m, the typical SI for forecast lead time of 3–15 h is 0.1–0.15 [36]. For the Mediterranean Sea with average SWH, ~2 m SI for forecast lead time 3–72 h is ~0.2–0.25 [35]. If we have a wave model of RMSE around 0.3 m and a satellite error of 0.3 m with a small average of SWH ~1 m, we will obtain a larger SI. In our case, the R correlation coefficient is 0.6–0.7 in summer months, which is also caused by a small average SWH. The wind speed is 2–4 m/s in summer (often due to a local mesoscale process). However, for our meteorological models, a correct wind forecast is challenging in this near calm condition. Thus, we have a low correlation for SWH.

COSMO forecast data with a high spatial resolution were used for wind forcing, but this did not lead to an improvement in the SWH forecast (Table 3). Mesoscale meteorological models that allow describing the underlying surface with a higher resolution use more advanced parametrizations for the surface layer compared to global models. Analysis of the results presented in Tables 1 and 3 shows that, according to RMSE and SI, the best result is obtained by the WW3-GFS version for almost all forecast lead times. According to the R correlation coefficient, version WW3-COSMO is slightly better than other versions for the lead times of 3–39 h, but for the lead times 51–63 h, WW3-GFS is better again. Thus, in our case, COSMO forcing (with a better spatial resolution of 7 km) does not give an unambiguously better result than GFS with a step of 25\12 km. The surface roughness is not so important above the water. Furthermore, the waves have a uniformity scale of about 10 km (that is, the wave height changes little at this scale). So, an increase in spatial resolution does not improve the forecast quality. It is possible that, at the early stages of the forecast, the high-resolution model assimilates satellite wind data better; however, with an increase in lead time, the stability of the model deteriorates. On the maps of the spatial distribution of Bias (Figure 10), the WW3-COSMO model is statistically better only in the region of Novorossiysk where the local wind of Novorossiyskaya Bora [37,38] blows, which is reproduced with less accuracy by the WW3-GFS model.

Several additional difficulties arise when the quality of wave forecasts is assessed, in comparison with the wave hindcast. One of the problems here is that weather forecast data are usually not stored for a long time. The same is true for meteorological and wave

forecasts. Operational forecasts are stored for a week or a month and then erased. Consequently, either quality assessments must be carried out continuously, or forecast should be saved for a long time. It is also difficult to reproduce forecasts in the past if the weather forecast has not been saved. There are no such problems with wind reanalysis data usage. Reanalysis is always available. Wind wave forecast systems are estimated by authors as the hindcast version of the model usually. Due to such estimates, forecasts are used for industry, although there is sometimes an analysis of specific storm cases additionally. Only one article showed the quality for each lead time, although the forecast for 2–3 days is the most interesting to consumers [36,39].

The most important problem that we faced in this work is that even 1-year data give only a few storm cases (SWH > 4 m), recorded by satellites. Saral Altika tracks over the Black Sea twice per day and storms in the Black Sea develop and die out very quickly, though not all of them are registered by satellites.

The evaluation criteria are difficult to design for assessing the quality of forecasts. Standard approaches offer to compare observed value at the observation point with the forecast at this point (collocation) at the same moment (±1 h, for example). Let us suppose that the forecast lead time is 3 days and the storm waves are predicted but located away (50 km) from the point and with a time shift of 3–5 h. Is such a forecast accurate enough? Formally, the errors will be great, but in fact, the storm prediction already offers a benefit. Unfortunately, no clear criteria have been developed yet for assessing wave forecasts. The authors used the classical approach in this work. In the future, a new methodology for assessing the quality of forecasts with wider deltas (shifting) in time and space is planned to be used.

## 5. Conclusions

The quality of different versions of wind wave forecast systems in the Black Sea is carried out. The authors tested the forecasts based on the WAVEWATCH III and SWAN models using GFS and COSMO-RU07 forcing. Comparative analysis of the forecasts and the satellite measurements shows that statistical quality indicators of the models are quite satisfactory for modern wave models (correlation coefficient ~0.9–0.8, RMSE ~0.3 m). According to the RMSE and SI, the best result is shown by the WW3-GFS version for all forecast lead times. According to the R correlation coefficient, for the lead times of 3–39 h, version WW3-COSMO is slightly better than other versions; however, for the lead times of 51–63 h, WW3-GFS is better again.

A clear improvement of the wave forecast quality with high-resolution wind forecast COSMO-RU07 is not registered.

Low quality of wind and wave forecasts is observed in the summer months when there were low wind speeds and weak wind waves.

The Bias does not exceed 0.5 m in the SWH range from 0 to 3 m. However, the Bias sharply increases to −2 or −3 m for the SWH range of 3–4 m. However, this result is statistically insignificant because, for SWH, more than 3 m have only few storms in total.

The analysis of the two most severe storms with SWH of more than 4 m shows that wave forecast can overestimate or underestimate the wave height. Thus, the statistical series for high waves contains very few values and it makes a great problem for quality estimates.

The obtained results about wave forecast quality can be used in different industries: navigation, tourist activities, rescue operations, marine fishing, and others. Wave forecast is useful for the planning of power generation and for protecting the converters from waves that are too big.

**Author Contributions:** The concept of the study was jointly developed by A.Z., Y.R. and S.M. S.M. did numerical simulations by SWAN model, analysis, visualization, and manuscript writing. A.Z. did numerical simulations by WW3 model, analysis, visualization, and manuscript writing. Y.R. did manuscript writing. V.A. did numerical simulations by SWAN model. K.S.—analysis of numerical

modeling and manuscript writing. S.M. prepared the paper with contributions from A.Z., Y.R., K.S. and V.A. All authors have read and agreed to the published version of the manuscript.

**Funding:** The forecast of wave parameters using the SWAN model was made by S.A Myslenkov with the financial support of the Russian Foundation for Basic Research (RFBR) in the framework of the scientific project No. 18-05-80088. Data analysis funded by the Ministry of Science and Higher Education of Russia, theme 0128-2021-0002 (K.P Silvestrova). Implementation of the wave model for Black Sea was carried out by V.S. Arkhipkin with the financial support of RFBR in the framework of the scientific project No. 20-55-46007.

**Institutional Review Board Statement:** Not applicable.

**Informed Consent Statement:** Not applicable.

**Data Availability Statement:** Not applicable.

**Acknowledgments:** Authors gratefully thank E.V. Stoliarova for the constructive comments and recommendations.

**Conflicts of Interest:** The authors declare no conflict of interest.

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
