# Peer review of "Quality of the Wind Wave Forecast in the Black Sea Including Storm Wave Analysis"

_sustainability, doi:10.3390/su132313099_

Round 1
Reviewer 1 Report
The wave forecast model produced by the WAVEWATCH III model with COSMO-RU07
is one of interests for climate scientists. But the present paper lacks of some evidences to explain the performance of COSMO-RU07. The reviewer recommends for publication after revising the issues as followings:
1. Authors need to compare the wave and wind parameters between COSMO-RU07 and observations in temporal and spatial domain. Authors only provides one introduction of COSMO model which is applied to wave forecast model. Can authors describe more of the supporting COSMO model? e.g., Rivin [2015].
2. Wave forecast is also essential in temporal sequences to assess the extreme wave for a long-term duration. Authors can do time series comparisons with observations which could possibly be accessed from Onea and Rusu [2017] and Islek [2021].
3. What is the difference between the lead time of 15 and 39 hours ? Is any reason to support that the presented model is well-calibrated for the lead time of 15hours? If authors compare the buoy observation, the leading time is likely to be changed.
4. Throughout : Reference check for “.....atmospheric models, such as the COSMO-RU07 model [Rivin et al., 2011].” vs [Rivin et al., 2012].
5. Table 3. is missing the “-COSMO” in the column of R (correlation coef).
6. Throughout : format for “Scatter index” vs “SI”; format for “BIAS” vs “Bias”
7. Throughout : Delete the “-” in “….N-is….”
8. Line 101: Accroding to “... for the lead times 3-39 hours version WW3-COSMO is slightly better of other versions”, authors only show the WW3-GFS results during the strong wave condition. How about the results of other two models during the same period ? Do authors imply that the WW3-GFS model is much better performance than the others ?
9. Line 104: “A clear improvement of the wave forecast quality with high-resolution wind forecast COSMO-RU07 is not registered.” Can authors discuss more about the improvement by using COSMO-RU07 as model input ? Comparisons of three models are not well-interpreted and confused.
10. Line numbers are missing in page 1-17.
References
Islek, F., Yuksel, Y., Sahin, C., & Guner, H. A. A. (2021). Long-term analysis of extreme wave characteristics based on the SWAN hindcasts over the Black Sea using two different wind fields. Dynamics of Atmospheres and Oceans, 94, 101165.
Onea, F., & Rusu, L. (2017). A long-term assessment of the Black Sea wave climate. Sustainability, 9(10), 1875.
Rivin, G. S., Rozinkina, I. A., Bagrov, A. N., & Blinov, D. V. (2012). The COSMO-Ru7 Mesoscale Model and the Results of Its Operational Testing. Informational Collected Papers “The Results of Testing New and Improved Technologies, Models, and Methods of Hydrometeorological Forecasting, (39).
Rivin, G. S., Rozinkina, I. A., Vil’Fand, R. M., Alferov, D. Y., Astakhova, E. D., Blinov, D. V., ... & Chumakov, M. M. (2015). The COSMO-Ru system of nonhydrostatic mesoscale short-range weather forecasting of the Hydrometcenter of Russia: The second stage of implementation and development. Russian Meteorology and hydrology, 40(6), 400-410.
Author Response
Authors are very grateful to all reviewers for the hard work on article.
- Authors need to compare the wave and wind parameters between COSMO-RU07 and observations in temporal and spatial domain. Authors only provides one introduction of COSMO model which is applied to wave forecast model. Can authors describe more of the supporting COSMO model? e.g., Rivin [2015].
There are temporal estimates in Table 2 and Fig. 6. We also add the maps of the Bias spatial distribution for the wave height (Figure 10). There is a significant underestimation for the implementation of the GFS in the region of Novorossiysk. Apparently, the Novorossiysk bora is being reproduced in the COSMO realization [Alpers et al. 2010, Toropov et al. 2012]. Description of the COSMO model has been added.
- Wave forecast is also essential in temporal sequences to assess the extreme wave for a long-term duration. Authors can do time series comparisons with observations which could possibly be accessed from Onea and Rusu [2017] and Islek [2021].
Thanks for your advice, we found the articles. We understand the essence of such a comparison. We will request the data from the authors, but this will take additional time. Unfortunately, we don’t’ have the opportunity to make such a comparison now.
- What is the difference between the lead time of 15 and 39 hours ? Is any reason to support that the presented model is well-calibrated for the lead time of 15hours? If authors compare the buoy observation, the leading time is likely to be changed.
We have shown that the simulation quality of the wind speed and wave height deteriorates with increasing forecast lead time from 15 to 39 hours (selected as an example). Tables 1 and 3 show that the quality of the forecast for 15 hours is better than for 39 hours and beyond. The model is well calibrated for 0 hours. That is hindcast. Then the wind forecast starts to accumulate error, and hence the error in the wave height forecast also increases. Comparison with the buoy data will give the same result.
- Throughout : Reference check for “.....atmospheric models, such as the COSMO-RU07 model [Rivin et al., 2011].” vs [Rivin et al., 2012].
Corrected.
- Table 3. is missing the “-COSMO” in the column of R (correlation coef).
Corrected.
- Throughout : format for “Scatter index” vs “SI”; format for “BIAS” vs “Bias”
Corrected.
- Throughout : Delete the “-” in “….N-is….”
Corrected.
- Line 101: Accroding to “... for the lead times 3-39 hours version WW3-COSMO is slightly better of other versions”, authors only show the WW3-GFS results during the strong wave condition. How about the results of other two models during the same period? Do authors imply that the WW3-GFS model is much better performance than the others ?
In section 3.3 we show the quality for various wave height ranges, including storms with SWH > 3.5 m. The WW3-GFS version is better than the others. Also, we’ve added fig 14b, which also confirms that WW3-GFS for one of the severe storms has better quality than WW3-COSMO.
- Line 104: “A clear improvement of the wave forecast quality with high-resolution wind forecast COSMO-RU07 is not registered.” Can authors discuss more about the improvement by using COSMO-RU07 as model input ? Comparisons of three models are not well-interpreted and confused.
Add some new text about this in the Discussion.
COSMO forecast data with a high spatial resolution was used as a wind forcing, but this did not lead to an improvement in the SWH forecast (Table 3). Mesoscale meteorological models allow describing the underlying surface with a higher resolution, use more advanced parametrizations for the surface layer than global models. Analysis of the results presented in Tables 1 and 3 shows that according to RMSE and SI the best result is obtained by the WW3-GFS version for almost all forecast lead times. According to the R correlation coefficient, version WW3-COSMO is slightly better than other versions for the lead times 3-39 hours, but for the lead times 51-63 hours, WW3-GFS is better again. Thus, in our case, COSMO forcing (with a better spatial resolution of 7 km) does not give an unambiguously better result than GFS with a step of 25\12 km. Apparently, different underlying surface is not so important above the water, furthermore, the waves have a uniformity scale of about 10 km (that is, the wave height changes little at this scale). So, an increase in spatial resolution doesn’t improve the forecast quality. It is possible that at the early stages of the forecast, the high-resolution model assimilates satellite wind data better; however, with an increase in the lead time, the stability of the model deteriorates. On the maps of the spatial distribution of Bias error (Figure 10), the WW3-COSMO model is clearly statistically better only in the region of Novorossiysk, where the local wind of Novorossiyskaya Bora (Alpers et al. 2010, Toropov et al. 2012) blows, which is reproduced worse in the WW3-GFS model.
Reviewer 2 Report
In this paper the authors compare different approaches and numerical models using different wind forcing on different computational grids and models. Although the article is interesting and may be significant from a forecasting and numerical point of view, it is not possible to make an objective comparison and validation on performance, because different models, different grids and different forcing do not allow the comparison of results. I suggest broadening the analysis by forcing the models with the same forcing and evaluating the performance in every aspect so as to isolate the benefits and weaknesses of each approach. In addition, there are many problems in the writing of the test and in the description of the images, often very poor. Furthermore, the bibliography is very small, while in the literature there are many works on these topics.
1) In many part of paper we can find incomplete sentence such as: "
[NCEP GFS ..."
The Global Forecast System ...]
etc. Please comment and add reference.
2) "Improving the quality of meteorological forecasts can be achieved with increasing spatial resolution, in particular, using products of mesoscale atmospheric models, such as the COSMO-RU07 model [Rivin et al., 2011]"
add citation : Barbariol et al 2018
3) "But the wind speed from NCEP/CFSR reanalysis data is overes- timated according to satellite observation data with an average systematic error of 1 m/s.."
remove one dot.
4)
Author Response
Authors are very grateful to all reviewers for the hard work on article.
We understand that the study could be broadened, but still, the wind waves have a certain radius of homogeneity - the scale at which the wave parameters in space change little. Considering that this scale is about 10 km for the Black sea, we believe that differences in the step of the computational grid should not change the result much. Versions of WavewatchIII are based on the same grid and the same model parameters. The difference includes only wind forcing, therefore, in this part, the analysis should be considered objective.
1) In many part of paper we can find incomplete sentence such as: "
[NCEP GFS ..." The Global Forecast System ...]
These are references to number 13 and 14 in reference list. Corrected.
2) "Improving the quality of meteorological forecasts can be achieved with increasing spatial resolution, in particular, using products of mesoscale atmospheric models, such as the COSMO-RU07 model [Rivin et al., 2011]" add citation : Barbariol et al 2018
Could you please clarify which article do you mean?
3) "But the wind speed from NCEP/CFSR reanalysis data is overes- timated according to satellite observation data with an average systematic error of 1 m/s.." remove one dot.
Corrected.
Reviewer 3 Report
This manuscript compared a few wind data sets over Black Sea and highlighted the best one among them. The approach is well founded, but the analysis method is not promising. I have the following observations:
- Please provide line numbers. First 2 paragraphs in the introduction need references.
- 1st paragraph in the introduction: "We need to study wind waves conditions to prevent such disasters and high quality of wave forecast will certainly contribute to the sustainable development of the region." Why do you need it? The gaps, background information and motivations are not well explained before defining this objective.
- What are the full forms of "WAVEWATCH III "; "NCEP/NOAA "; "SWAN"; "COSMO-RU07"? Not everyone knows the full forms. Same comment for the entire manuscript.
- Overall introduction is not well organized. Objectives are not well defined with the gaps and motivations.
- Method section: The method used in this manuscript is very simple and not promising.
- Results Section: Whether the results are statistically significant? You didn’t also discuss the uncertainties in the wind wave forecast over Black sea.
- 1st paragraph, Result section: "As a result, wave height". It is not clear. Please rewrite the 1st sentence.
Author Response
- Please provide line numbers. First 2 paragraphs in the introduction need references.
We are very sorry, but in this template, we couldn’t insert line numbers.
References were added.
2. 1st paragraph in the introduction: "We need to study wind waves conditions to prevent such disasters and high quality of wave forecast will certainly contribute to the sustainable development of the region." Why do you need it? The gaps, background information and motivations are not well explained before defining this objective.
We add some new text in the Introduction. Thank you for the comment.
3. What are the full forms of "WAVEWATCH III "; "NCEP/NOAA "; "SWAN"; "COSMO-RU07"? Not everyone knows the full forms. Same comment for the entire manuscript.
Corrected
5. Method section: The method used in this manuscript is very simple and not promising.
The article is already overloaded, methods of wave modeling are described many times in other articles, but if the reviewer insists, then we can add a full description of the models.
6. Results Section: Whether the results are statistically significant? You didn’t also discuss the uncertainties in the wind wave forecast over Black sea.
In section 3.3 we describe the sample length for different wave height ranges. the results are undoubtedly significant for wave heights up to 2.5-3 m, then there are problems with insufficient data.
7. 1st paragraph, Result section: "As a result, wave height". It is not clear. Please rewrite the 1st sentence.
Corrected
Round 2
Reviewer 1 Report
The current manuscript is well revised and also clarified some issues of wave forecasting results.
Author Response
Thank you very much for your work with our manuscript!
Some sections have been further improved based on remarks from another reviewer.
Reviewer 3 Report
I appreciate authors' effort for considering my comments and incorporating some of them in the revised manuscript. However, I did not find the revision is conducted properly. I confirm authors have improved the introduction section to some extends, but other sections remained untouched. Authors did not perform any statistical tests also. I believe authors will improve the results section and provide some uncertainty information while submitting their final version.
Author Response
The authors are very grateful to the referee for important comments.
- We add "Table 5 Fisher's transformation of the SWH correlation coefficient...." where we provide the assessments of the significance of the correlation coefficients. Also description text added
- In section 3.2 we add the comments about the Bias error over the Black Sea.
- A small discussion about the distribution of bias error over the Black Sea is present in section 4 "discussion".